# Correlations between Hotel Size and Gas Consumption with a Feasibility Analysis of a Fuel Switch—A Coastal Case Study Croatia Adriatic

**Maja Štimac [1], Mario Matković [2] and Daria Karasalihović Sedlar [1,*]**

[1] Faculty of Mining, Geology, and Petroleum Engineering, University of Zagreb, 10000 Zagreb, Croatia; mstimac@rgn.hr

[2] Hilton Imperial Dubrovnik, 20000 Dubrovnik, Croatia; mario.matkovic@hilton.com

* Correspondence: daria.karasalihovic-sedlar@rgn.unizg.hr

**Abstract:** This paper analyses gas consumption in hotels on the Adriatic coast, comparing data on natural gas and liquefied petroleum gas (LPG) consumption by hotel size. The research hypothesis is that by switching from LPG to natural gas, not only can a reduction in emissions be achieved in the hospitality industry, but there are also significant economic benefits. The research objectives included a regression analysis for various factors affecting gas consumption. The analysis showed a medium–strong relationship between the variables, which is a novelty for energy trends in the hospitality sector. By converting from heating oil to natural gas, hotels can achieve significant financial savings. It was also calculated that this would reduce the total energy consumption costs. Measures taken by the hospitality industry will have a positive impact on guest perception and could be used as a promotional tool under the "green hospitality" label.

**Keywords:** energy transition; LPG; gas consumption; green hospitality; sustainability; decarbonisation; fuel switch





## 1. Introduction

To lessen the damaging effects of tourism on the environment, the tourist industry, similarly to all other sectors, is undergoing an energy transition from traditional fossil fuels to less carbon-intensive fuels. Many references claim that tourism significantly contributes to energy use and greenhouse gas emissions. The development of tourism substantially impacts environment quality, predominantly transportation, energy, and carbon-intensive commodities. Therefore, tourism is an important factor in climate change [1,2]. Previous research has mainly focused on the fact that tourism-related activities such as transportation, accommodation, etc., heavily depend on energy consumption that produces $CO_2$ emissions [3]; this means that tourism can become more sustainable by reducing its carbon footprint during the energy transition [4].

Tourism contributes significantly to global greenhouse gas (GHG) emissions, generating about 8% of global GHG emissions [5]. A large proportion of these emissions come from hotel units [6]. According to Salehi et al. [7], currently hotels hold a share of 21% in this carbon footprint [8], but it is projected that by 2035 it will increase to 25% [9]. In the service sector, hospitality is known for its tremendous environmental impact since hotels operate 24 h daily (including weekends) compared to the other service segments [10]. The hotel sector accounts for 1% of total global GHG emissions [11].

Adopting renewable energy sources such as solar, wind, and hydroelectric power is one strategy to achieve energy transformation in tourism [12], but implementing renewable technologies could impose expensive costs; in additions to this, the industry faced unprecedented challenges and threats from the impact of the COVID-19 pandemic [13,14]. Therefore, there is the necessity for an inclusive, bottom-up strategy for energy transition

and sustainability-led innovations in the hospitality industry [15]. Instead of imposing expensive imported technologies, this strategy encourages local and cheaper technology implementation to ease the transition. Utilizing energy-efficient technologies is another strategy for the energy shift in tourism, along with greening performance in the hospitality industry [16]. Investment in the hospitality industry is a crucial decision [17], and hotels need to invest constantly in financing capacity expansion, growth, and renovation [18–20], but also energy.

The tourism and hospitality industry is a critical economic sector and provides countries and regions with economic growth and economic and social development [21]. Government policy has a considerable influence on the hospitality industry's history [22], and it is also expected to have an encouraging role in energy transition in tourism by using policy tools that include feed-in tariffs, tax incentives, and subsidies. For example, one of the practical policies could be the implementation of green fiscal policies, green construction in hotels and tourist accommodation, green globalization, etc. [23].

However, there are obstacles to the fuel switch in the tourism industry. The high cost of renewable energy technology is one of the main problems [24]. Moreover, previously the tourism sector has not reported any benefits from investing in renewable energy sources, efficient energy production, or reduced energy costs [25,26]. The major obstacle to renewable energy technology adoption is often its high cost. It is also difficult to integrate renewable energy technologies in the tourism sector because of a lack of awareness and information.

Based on the intensive carbon footprint [27] and operative cost intensiveness [28] of hotels, the research hypothesis has been set that by switching from LPG to natural gas, not only can a reduction in emissions be achieved in the hospitality industry but there are also significant economic benefits.

As hotels in China are the highest energy consumers among public buildings after shopping malls [29], it is reasonable to expect that rising environmental concerns will soon affect the hospitality sector. Energy management must be considered a significant part of overall environmental management due to the financial savings that can be achieved [30]. Energy management in hotels has been organized similarly to water management, despite its greater complexity [31]. Moreover, hoteliers tend to incorporate some form of waste management, and when it comes to energy-saving measures, the most popular is energy-efficient lighting [32]. The other significant energy-saving measures are wall insulation, window substitution, and the installation of heat pumps [33]. Other cost-cutting methods, such as less regular linen and towel changes, reducing the usage of air conditioning, and investing in upgrading energy systems, are not as common [34]. As environmental awareness increased, more energy-saving equipment and products were adopted, and air conditioning rationalization became the most important measure to reduce the environmental footprint [35]. For hotels that take environmental impacts more seriously, there are additional considerations that can be taken into account from the beginning of the hotel construction, such as the orientation and insulation of the building, rooms, and windows and the installation of mechanical ventilation [36]. A comprehensive review paper [37] uncovered a debate on whether a business achieves savings through operational efficiencies or whether green practices increase time and cost to the point where profit from the core business decreases. This revealed a research gap in determining a relationship between a hotel's financial performance and environmental responsibility. Prominent tourism destinations such as Hawaii have already recognized the importance of improving energy efficiency, and hotels have significantly reduced their electricity and fuel consumption thanks to a wave of renovations in the early to mid-2000s [38]. This research also confirmed the usefulness of investigating this issue further.

Previous studies, such as those by Tsai et al. [39], Wang [40], and Warnken et al. [41], have shown that hotels that offer higher-quality service consume more energy [42]. In terms of making the hospitality industry more environmentally friendly, the next step towards a greener and more sustainable industry is certainly the transition to cleaner fuels.

Low-carbon development is an important measure to reduce environmental degradation due to increased economic activity [43–46]. Recent studies [47] have shown that reducing carbon emissions increases the productivity and brand value of the hotel chains studied. Since natural gas is recognized as a cleaner-burning fuel compared to coal and oil [48], it is a logical choice wherever the appropriate infrastructure is available. While great efforts are being invested into promoting renewable energy sources, the low prices of fossil fuels still make them attractive, and they play an important role in the energy mix [49], with their share in primary energy consumption gradually decreasing.

In this paper, gas consumption in hotel accommodation is discussed, categorizing hotels according to the type of gas used and operating characteristics during the year—seasonal and non-seasonal hotels. The literature review has shown that no similar analysis has been previously conducted in other regions, and this type of analysis has been proposed based on data availability. A total of 70 hotels on the Croatian Adriatic coast were surveyed, with 51 hotels included in the analysis of gas consumption data. Even though this is only 12% of the total number of hotels [50], the analysed data are more than representative since they include all the biggest hotels and hotel chains in the researched area. Extra small-type hotels with only a few rooms have not been analysed. A regression analysis was carried out to determine the relationships between the main parameters of hotel size and a hotel's natural gas consumption to categorize the hotels, which will eventually allow further analysis regarding natural gas consumption and any related benefits, which are certainly dependent on hotel size. Tang et al. [51] analysed hotel energy consumption in Lijang (China) based on a regression-based model. Oluseyi et al. [52] analysed 28 hotels in Lagos (Nigeria) where energy consumption per room was calculated. Furthermore, based on the hypothesis that, in addition to the environmental benefits resulting from emission reduction, switching to a cleaner fuel would also lead to financial savings, data were collected for a hotel that switched from heating oil to natural gas, and then a cost comparison was made for a hypothetical case of switching from LPG to natural gas. The hotels considered in this study are all located on the Croatian coast. This region is of significant interest due to 92.5% of all Croatian accommodation capacity in 2019 being in this region [53]. To take full advantage of a switch to a cleaner fuel, it is necessary to upgrade the current gas infrastructure, as most of the Dalmatian region does not have access to the gas transmission and distribution network [54].

## 2. Materials and Methods

It is believed that a significant amount of the energy consumed in hotels is wasted, leaving much room for improving energy efficiency and resource conservation [29]. Bianco et al. [33] analysed energy-saving opportunities in the Italian hotel sector, as hotels offer a wide range of services to their customers and are therefore among the building types with the highest energy consumption. The bottom-up model was developed for estimating energy consumption, namely electricity and natural gas, in the Italian hotel sector. Shao et al. [55] analysed the electricity data of hotel buildings using a data prediction model to achieve appropriate energy consumption and save energy. Bianco [56] analysed factors affecting electricity consumption in the tourism sector in Italy. The same framework can be applied to any other country and energy source. The study by Pablo-Romero et al. [42] focused on the relationship between tourism growth and energy consumption in the hospitality sector in the Mediterranean Spanish provinces.

The increasing number of publications on energy efficiency in hotels [33,57,58] is nothing but proof that energy management has become an important issue in the industry, not only because of the direct financial savings and protection of the environment but also because of the positive impact that the implementation of green practices has on the reputation of a hotel. The trend to reduce carbon emissions and growing environmental awareness have made green hotels an important business direction for the hospitality industry [59]. The impact of climate change is mitigated by reducing carbon emissions and promoting global sustainability through implementing green practices [60]. According

to Arenhart et al. [61], hotel chains can contribute to a more sustainable future of their operations by investing in updating their systems and therefore building positive marketing for their brands, retaining a greater number of guests, and generating a long-term financial return.

In 2018, the EU adopted Directive 2018/844/EU [62] on the improvement of the energy performance of buildings, partially amending Directive 2010/31/EU on the energy performance of buildings [63]. Directive 2018/844/EU [62] also underlines the importance of improving the energy performance of current and future buildings. One of the most important areas of application of these regulations is tourist hotel buildings, as tourism has a significant impact on society [64]. Tourism globally represents 9% of the world's GDP, with a revenue of more than USD 1400 billion [65] and a constant income increase of 195% over the last 20 years, creating 17.6% of global jobs [66].

Today, tourism is one of the pillars of the Croatian economy and thus has a noticeable impact on improving Gross Domestic Product (GDP) [67]. Tourism's direct and indirect effects on the economy were estimated at almost 20% of GDP and 24% of total employment in 2019 [68,69]. In Croatia, hotels account for a large part of tourist accommodation, so it makes sense to focus on their energy efficiency. Hotels' energy consumption and emissions can vary depending on the geographical region, facilities, and categories of establishments [66]. The hotel industry in Croatia is dominated by large establishments in the low–mid price range [70]. Energy consumption in hotels includes electricity, natural gas, liquefied petroleum gas (LPG), but also renewable energy sources. Gas is used in hotels for a variety of purposes, most commonly for space heating, heating water in swimming pools, and preparing hot water in restaurant kitchens [71]. In Croatia, natural gas accounts for about 28% of primary energy consumption in the service sector [72]. An increase in this percentage is expected after more branches are added to the existing gas network, and the breadth of gas use will guarantee that the positive environmental impact is measurable. Although LPG is a significant improvement over heating oil, the use of natural gas is an even greater step forward from an environmental perspective. This is mainly because the "dirty" component of a hydrocarbon is carbon. In natural gas, there is a more favourable ratio of hydrogen to carbon, as it consists mainly of methane [73], while LPG can be sold in different proportions of a mixture of propane and butane, depending on customer requirements [74].

The gasification of the region is a viable option that will help reduce $CO_2$ emissions as it would increase the share of the cleanest fossil fuel in the hotels' energy mix. This would have a direct impact on overall emissions as hotels are the main energy consumers in the service industry [75]. Due to their size, hotels can be considered frontrunners in decarbonizing the hospitality industry.

To study gas consumption in hotels on the Croatian coast, a questionnaire was prepared by us in 2022 to survey gas consumption. The main questions of the questionnaire are given below. The data on gas consumption in hotels on the Adriatic coast were collected during May 2022 in different ways: online survey, email, telephone, and in-person on-site. Most hotels responded to the survey questionnaire via email and included additional tables with their gas consumption data for five years, from 2016 to 2021. The research and survey included a total of 70 hotels on the coast, of which 51 were further analysed for gas consumption. The remaining hotels did not include their monthly gas consumption tables, or they provided incomplete data and were therefore excluded from the analysis. The main research questions of the survey were:

1.　What type of gas is used in a hotel facility? Liquefied petroleum gas/LNG/natural gas?
2.　What is the area of the hotel building in $m^2$?
3.　What is the accommodation capacity of the hotel?
4.　What was the average monthly gas consumption from 2016 to 2021?
5.　For what purposes is the gas used?
6.　When is the hotel open during the year?

With the obtained data on gas consumption, a database was created for each hotel chain. An analysis of the dependence of gas consumption on the total building floor area and the total number of rooms was carried out. The data included in the analysis were systematically processed in terms of the hotels' mode of operation, i.e., their classification as seasonal or non-seasonal. In the preparation of the charts, Microsoft Excel version 16.72 and linear regression were used by authors in Zagreb, Croatia, which gave the regression line and the corresponding equation, and the correlation coefficient $R^2$ was calculated.

### 2.1. Type of Gas

When asked what type of gas is used in the accommodation facility, hotels on the Adriatic coast are divided into two major regions: (1) Istria and Kvarner and (2) Dalmatia. About 70 hotels were surveyed in Istria and Kvarner, and the same number in Dalmatia.

Based on the survey results, it can be found that in the first region the representation of LPG and natural gas as the energy used is almost the same, which can be seen in Figure 1, but still, LPG is represented at a slightly higher percentage of 60%. Natural gas is represented at about 40%, and the main reason for this is the good gas connection of most of Istria. In Istria, in Pula in 2021, a number of the hotels that were not yet connected to the natural gas distribution network were connected, which reduced the use of LPG after the extension of the gas distribution network [76].

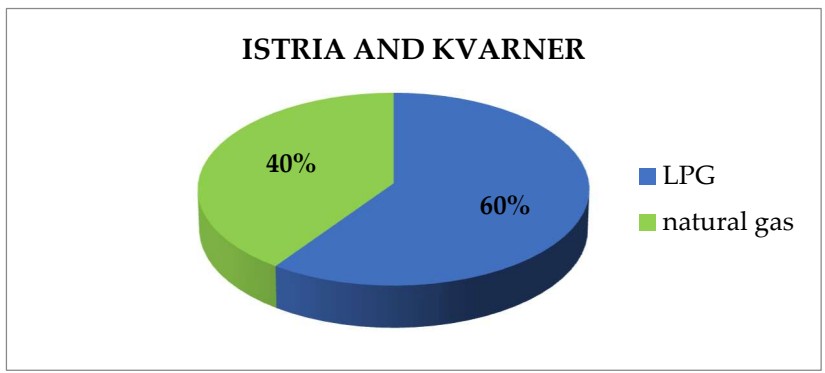

**Figure 1.** Graph of the representation of LPG and natural gas in Istria and Kvarner in hotels.

Based on the analysis of the data on gas consumption in Dalmatia (Figure 2), from a sample of about 70 hotels, it can be concluded that LPG is used to a greater extent (87%) compared to natural gas (13%). There is a greater difference in representation compared to Istria and Kvarner, which is due to the lack of a gas distribution network in most parts of the region. The natural gas market is dependent on a transport network, as natural gas is only available where a network exists [77].

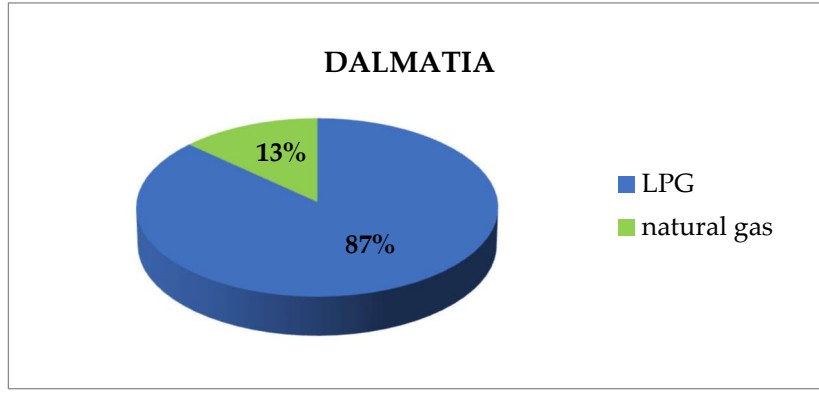

**Figure 2.** Graph of LPG and natural gas consumption in hotels in Dalmatia.

The gas transmission network extends only to Split, which prevents most of the Dalmatia region from connecting to the gas transmission system, while the distribution network is implemented only in a small part of the whole of Dalmatia. In these parts of the region, gas replaces electricity as the primary energy source. In addition, air-to-water and water-to-water heat pumps are used for heating and cooling. Heating oil is also used in smaller quantities, mostly in hotels that have not yet been renovated.

### 2.2. Dependence of Variables on Gas Consumption

To better analyse the collected data on gas consumption, it is necessary to determine its dependence on various factors such as the total area of the building, the total number of rooms, the size of the rooms, the number of nights, etc. For this purpose, regression analysis has been used.

Regression analysis is a statistical method that studies the relationship between variables. Here, linear regression is used to show the correlation of a variable $y$ to an independent variable $x$. In linear regression, it is assumed that the $x$ variable is a continuous random variable that depends linearly on the $y$ variable and is described by the regression line (Equation (1)):

$$y = a + bx \tag{1}$$

The square of the correlation coefficient, $R^2$, shows the quality of the relationship between the two variables [78]. If there is no dependence between variables $x$ and $y$, the correlation factor is 0, and $R^2$ increases with the degree of correlation to 1. The correlation factor $R^2$ is calculated by Equation (2):

$$R^2 = \frac{\sum_{i=1}^{n}(\hat{y}_i - \bar{y})^2}{\sum_{i=1}^{n}(y_i - \bar{y})^2}, \quad 0 \leq R^2 \leq 1 \tag{2}$$

where the variance due to the regression model ($SS_{err}$) is given by Equation (3) and the total variation ($SS_{tot}$) is represented by Equation (4):

$$SS_{err} = \sum_{i=1}^{n}(\hat{y}_i - \bar{y})^2 \tag{3}$$

$$SS_{tot} = \sum_{i=1}^{n}(y_i - \bar{y})^2 \tag{4}$$

$SS_{err}$ represents the percentage of variation in the values of $y$ that is not explained by the model, assuming that it is caused by random errors in the determination of the variable $y$. $SS_{tot}$ represents the total variation in the values of $y$ [78]. According to the Chaddock scale (Table 1), the dependence of the variables can be determined:

**Table 1.** Chaddock scale for $R^2$ value (adjusted according to [79]).

| $R^2$ | Interpretation |
| --- | --- |
| 0 | There is no correlation |
| 0.00–0.25 | Weak correlation |
| 0.25–0.64 | Medium correlation |
| 0.64–1 | Strong correlation |
| 1 | Full correlation |

### 2.3. Regression Analysis for Gas Consumption

Gas is used in hotels for various purposes: cooking in restaurants, heating the building, heating hot water, and heating water in swimming pools or laundries. When heating a building, for example, the total area of the building has a big impact on the amount of gas used, as does the insulation of the building and the efficiency of the building. Gas boilers are mainly used for heating. Some of the factors that affect gas consumption in hotels are:

- total floor area;
- number of rooms;
- room area;
- number of meals;
- heating/cooling (if gas is used);
- guest nights;
- occupancy rate.

In terms of variables affecting energy efficiency, Tang et al. [51] found that the number of guest rooms and the room revenue are among the key factors contributing to a hotel's energy efficiency. An analysis of the dependence of gas consumption on the total building floor area and the total number of rooms was carried out. For the remaining factors, it was not possible to perform an adequate regression analysis due to insufficient data. The data included in the analysis were systematically processed in terms of the hotels' mode of operation, i.e., their classification as seasonal or non-seasonal. In the preparation of the charts, Microsoft Excel and linear regression were used, which gave the regression line and the corresponding equation, and the correlation coefficient $R^2$ was calculated.

### 3. Results

#### 3.1. Regression Analysis of Gas Consumption for Non-Seasonal Hotels

The regression analysis showed that total gas consumption and total floor area of the building, total gas consumption, and the total number of rooms have a medium to strong relationship, according to the Chaddock scale. The regression coefficient $R^2$ for the dependence of annual gas consumption on the total floor area of the building is 0.7243. For the dependence of annual gas consumption in kWh on the total number of rooms, the regression coefficient is slightly lower and is 0.4458. Equations with an $R^2$ greater than 0.6 can be useful for calculating future gas consumption in hotels and other service sectors. Figures 3 and 4 show the obtained data graphically displayed using linear regression.

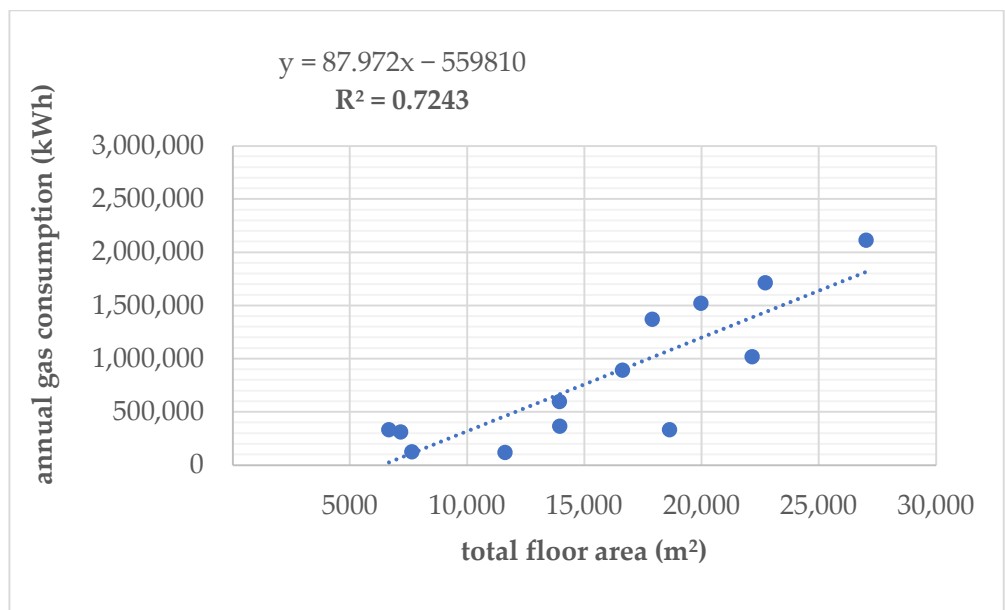

**Figure 3.** Graphical representation of the annual gas consumption of non-seasonal hotels in kWh to the total floor area in m$^2$.

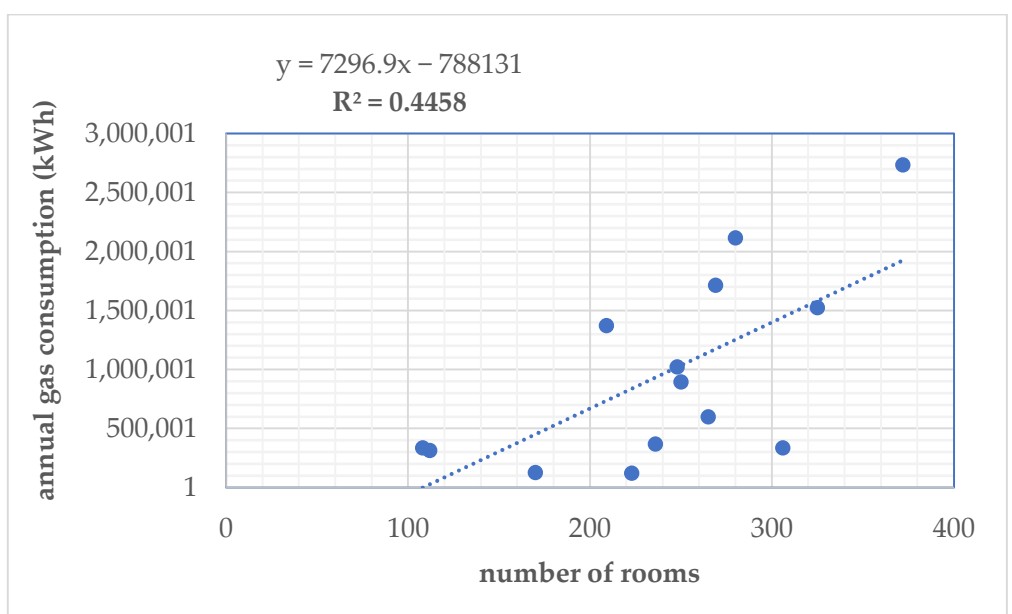

**Figure 4.** Graphical representation of annual gas consumption of non-seasonal hotels in kWh to the total number of rooms in the hotel.

The annual gas consumption of non-seasonal hotels in kWh per m² concerning the total building area and concerning the total number of rooms in the hotel shows no significant relationship, as can be seen in Figure 5. The regression coefficient $R^2$ is 0.3396 concerning the total number of rooms in the hotel, which is interpreted as a medium–weak relationship according to the Chaddock scale; there is no significant correlation.

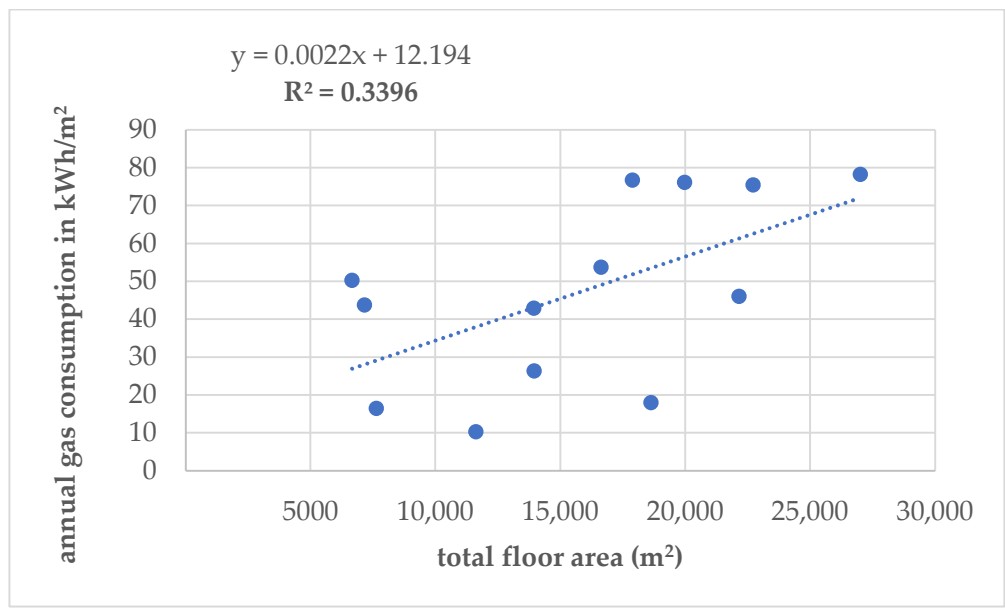

**Figure 5.** Graphical representation of annual gas consumption in non-seasonal hotels in kWh per square meter as a function to total floor area in m².

### 3.2. Regression Analysis of LPG Consumption for Non-Seasonal Hotels

The dependence of annual LPG consumption on the total number of rooms and the total area of the hotel building was considered. The analysis showed that in non-seasonal hotels there is no significant correlation between the above dependencies. The lack of correlation can also be seen in the graphs in Figures 6 and 7. This can be explained by

large discrepancies between the gas consumption data obtained and the hotel sizes even in the hotel sample used. When the annual LPG consumption in kWh is related to the total number of rooms, a weak correlation is obtained; the correlation coefficient is 0.1258. The same result was obtained when the LPG consumption was related to the total floor area of the building, where $R^2$ is 0.0911.

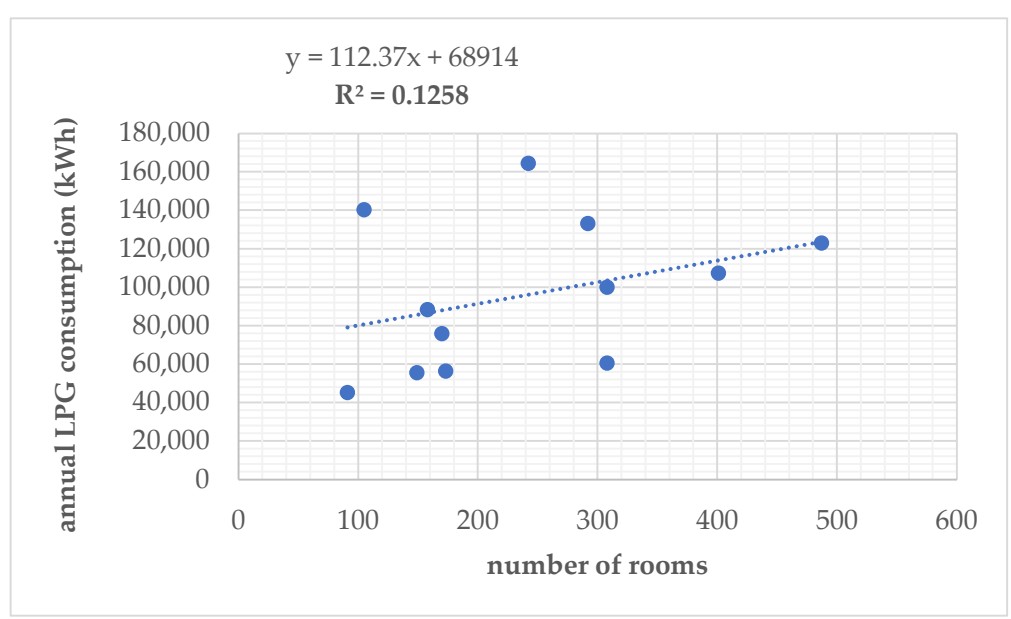

**Figure 6.** Graphical representation of annual LPG consumption in non-seasonal hotels in kWh vs. the total number of rooms in the hotel.

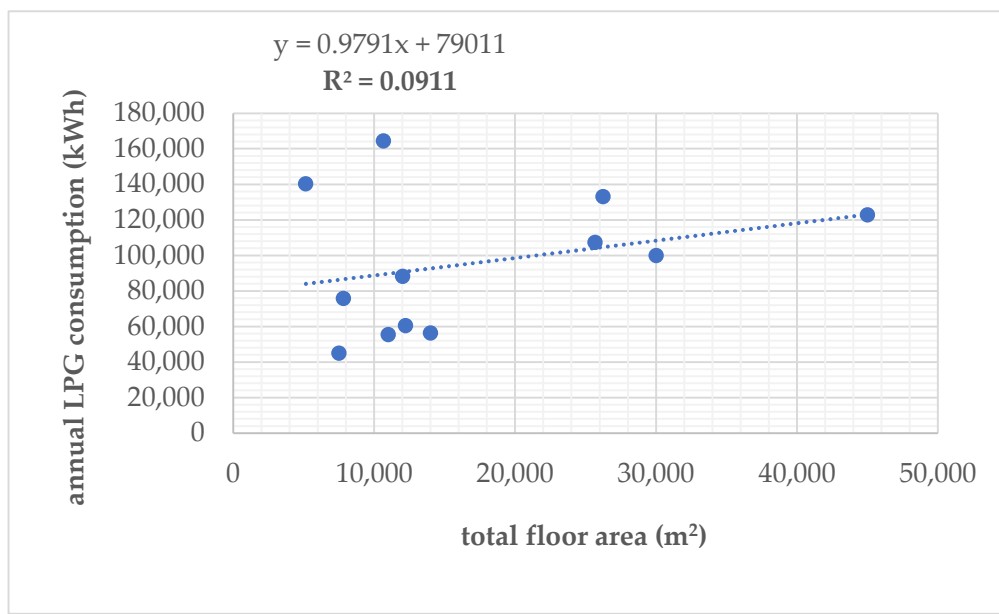

**Figure 7.** Graphical representation of annual LPG consumption in non-seasonal hotels in kWh concerning the total floor area of the building in m$^2$.

## 4. Discussion

### 4.1. Gas Consumption in Hotels

After systematizing the collected data, the hotels were divided into three categories according to their size: small, medium, and large hotels. The categorization itself was discretionary and based on the area of the hotel building. The hotel is categorized as small

for any lodging facility with an area of up to 10,000 m$^2$, medium for a lodging facility with an area of 10,000–20,000 m$^2$, and large for any lodging facility with an area of more than 20,000 m$^2$.

Figure 8 shows the average consumption of natural gas in hotels on the entire Adriatic coast for the past five years from 2016 to 2021. For greater accuracy in the analysis of the collected data, the average consumption of natural gas is presented for each hotel concerning the total area of the hotel building. The data are shown in kWh per m$^2$. It can be seen from the graph that the average monthly natural gas consumption in the category of small, medium, and large hotels is an approximation. On an annual basis, small hotels consume an average of 1.3 kWh of gas per m$^2$, medium hotels consume an average of 2.8 kWh of gas per m$^2$, and large hotels consume about 3.9 kWh per m$^2$ of natural gas. However, large hotels consume slightly more gas, which is related to the larger area to be heated. They need to heat large common areas such as restaurants, swimming pools, and conference rooms, which correlates with the movement of meteorological conditions throughout the year.

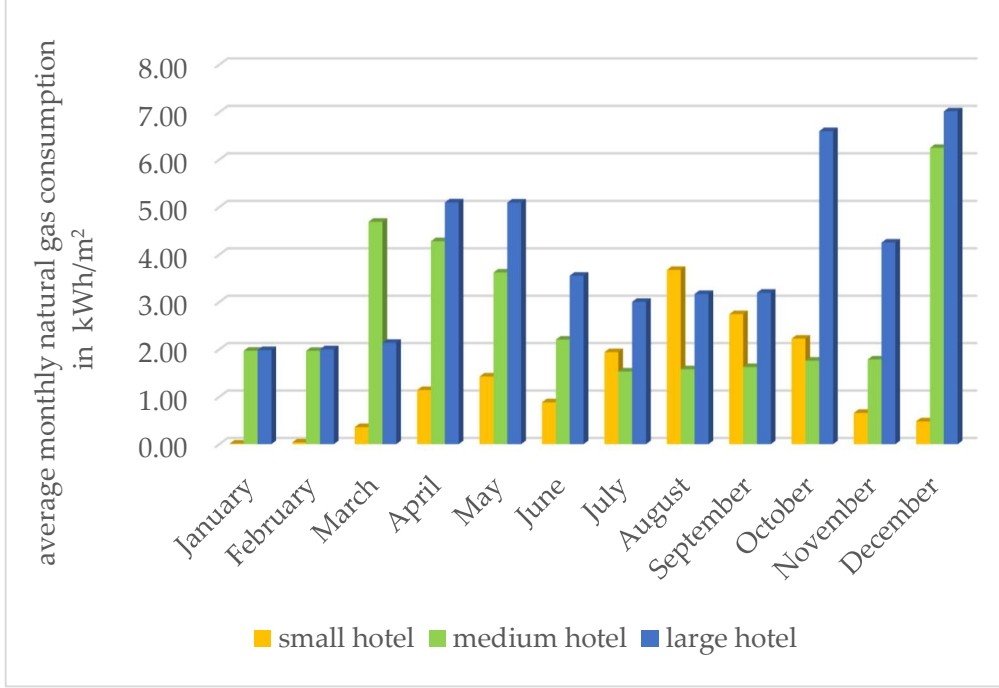

**Figure 8.** Graphical representation of average monthly natural gas consumption from 2016 to 2021 in kWh per m$^2$ in categorized hotels by size.

Figure 9 shows the dependence of average LPG consumption in kWh per m$^2$ in hotels categorized by size from 2016 to 2021. Liquefied petroleum gas is mainly used in hotels located on islands or the coast in areas where there is no gas network. For the most part, LPG is used exclusively for cooking in most medium and large hotels, while a smaller percentage is used for both heating and domestic hot water (DHW). Some hotels also use LPG for laundry. At the annual level, small hotels consume on average 1.17 kWh of gas per m$^2$, medium hotels about 0.56 kWh of gas per m$^2$, and large hotels 0.42 kWh of gas per m$^2$. Small hotels differ in this graph from the value of the average LPG consumption of medium and large hotels because more hotels in this category use LPG for building heating, water heating, and pool water. It is important to analyse the age of the hotel and the climatic conditions of each hotel to improve the correlation of the data.

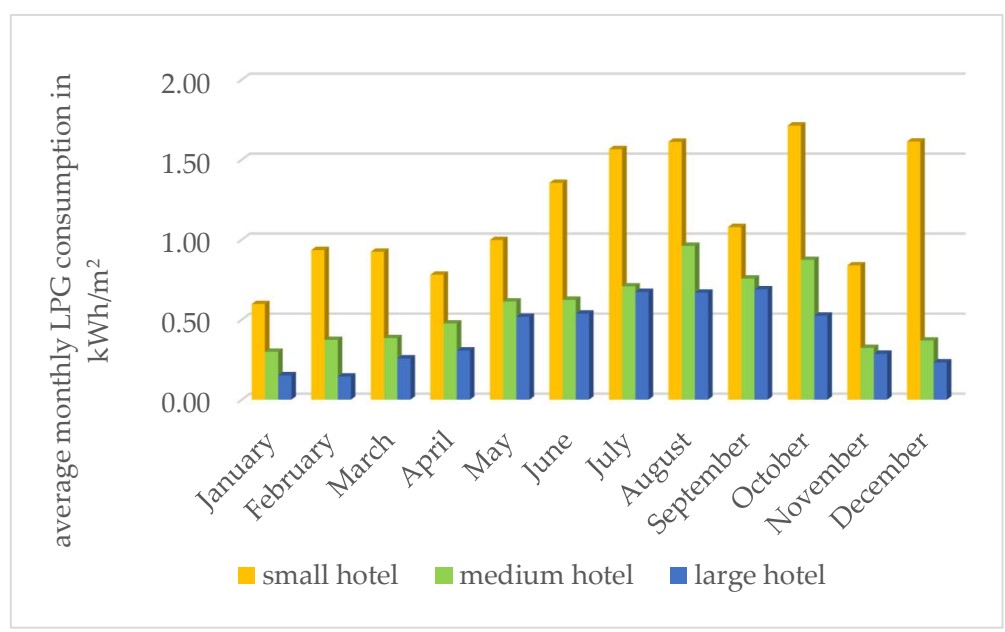

**Figure 9.** Graphical representation of average monthly LPG consumption in kWh per m² in hotels, categorized by size, from 2016 to 2021.

### 4.2. Conversion of Hotels from Heating Oil to Natural Gas

Below is a summary and comparison of the consumption of heating oil and natural gas in 2018 and 2019 for a hotel in Istria. In 2018, the analysed hotel used heating oil as its primary energy source and gas in tanks (LPG) for cooking. In 2019, the hotel switched from heating oil to natural gas; the costs were lower and amounted to EUR 21,868 for natural gas. Natural gas was used for heating the facility, swimming pool, and domestic hot water through the gas boiler room, and the kitchen is directly connected to the municipal gas distribution network. The data are presented in Tables 2 and 3. The period in which the hotel was open for guests, from April to the end of October for 2018 and 2019, was considered as a reference.

This analysis is relevant since it represents the main tourist region in Croatia along with Dalmatia, but Istria often has a leading role in investing in luxury hotel renovation, energy efficiency, and applying advanced technologies. The analysed hotel is comparable in size and location and could be representative since it used heating oil for heating the facility, swimming pool, and domestic hot water, as in the majority of analysed hotels, and it switched to natural gas. The other hotels have not yet switched to less carbon-intensive fuels or renewable sources of energy.

**Table 2.** Data on monthly consumption of heating oil in litres and LPG in kWh for one hotel in 2018.

| | 2018 | | | |
|---|---|---|---|---|
| **Month** | **Heating Oil** | | **LPG** | |
| | **L** | **EUR** | **kWh** | **EUR** |
| April | 7872 | 3913 | 10,798 | 470 |
| May | 19,180 | 10,091 | 13,394 | 581 |
| June | 9604 | 5395 | 12,804 | 556 |
| July | 9713 | 5381 | 12,844 | 518 |
| August | 18,491 | 10,293 | 13,789 | 555 |
| September | 9182 | 5295 | 11,950 | 481 |
| October | 7991 | 4787 | 10,100 | 408 |
| TOTAL | 82,033 | 45,155 | 85,679 | 3570 |

**Table 3.** Data on monthly consumption of heating oil in litres and natural gas in kWh for one hotel in 2019.

| | 2019 | | | |
|---|---|---|---|---|
| **Month** | **Heating Oil** | | **Natural Gas** | |
| | **L** | **EUR** | **kWh** | **EUR** |
| April | 0 | 0 | 113,364 | 4528 |
| May | 0 | 0 | 112,497 | 3917 |
| June | 0 | 0 | 81,853 | 3093 |
| July | 0 | 0 | 63,759 | 2407 |
| August | 0 | 0 | 81,635 | 3034 |
| September | 0 | 0 | 65,177 | 2464 |
| October | 0 | 0 | 66,352 | 2426 |
| TOTAL | 0 | 0 | 584,637 | 21,868 |

When calculating the savings after the conversion of hotels from one energy source to another, LPG was also considered, which was used for cooking in 2018, and the total cost of which amounted to EUR 3570. Accordingly, the total cost in 2018 was EUR 48,724. The total cost of natural gas consumption in 2019 was EUR 21,868. Finally, comparing the total cost from 2018 and the cost of natural gas from 2019, the savings are 55.12%. The savings are greater than 50% and show that, in addition to the environmental benefits, there is also an economic advantage to switching from using heating oil to using natural gas. The investment involved the conversion of the existing heating oil hot water boiler room into a gas boiler room. Previously, the hot water boiler room consisted of three hot water boiler units with a total heat output of 7 MW. Following the study and based on the analysis of available data on daily and seasonal heating oil consumption, the optimum heat output of 3 MW was estimated for the replacement of the gas appliances. The most investment-friendly option was the three-boiler design with a single-rated heat output of approximately 1 MW on the hotel roof. Most of the costs were for the gas roof-top heaters and the local gas and hot water installations. Other costs included preparatory works, the construction of heating pipes from the boiler room to the rooftop gas appliances, installation of gas connections, hydrant piping, insulation, etc. The total investment costs amounted to EUR 373,075.

Conversion to natural gas resulted in a decrease in consumption compared to the previously used energy source—heating oil.

*4.3. Conversion of the Hotel from LPG to Natural Gas*

The calculation of savings in the conversion of a hotel from LPG to natural gas was conducted in a situation where connection to a gas network was possible. For the consumption calculations, the average data of small category hotels using LPG for cooking, heating, and DHW were used. Table 4 shows the data on average LPG consumption in kilograms in 2022. The calculated data for the situation, if the same hotel were to switch to natural gas, are also included. The cost of LPG is calculated by multiplying the amount of gas consumed in kilograms by a tariff of 1.30 EUR/kg including VAT. This tariff is changed every week and a new LPG price list is sent to customers every Monday. To obtain the quantity of gas consumed in kWh, the determined quantity of gas consumed in kilograms is multiplied by the lower calorific value of 12.64 kWh/kg [80]. The cost of the calculated amount of potential natural gas consumption was obtained by multiplying the amount of consumed natural gas in kWh by the price of EUR 0.0530 including VAT [81].

**Table 4.** Data on LPG consumption in kg in 2022 and calculated data for natural gas in kWh.

| | 2022 | | Calculated Data | |
| --- | --- | --- | --- | --- |
| **Month** | **LPG** | | **Natural Gas** | |
| | **kg** | **EUR** | **kWh** | **EUR** |
| January | 114 | 148 | 1444 | 96 |
| February | 0 | 0 | 0 | 0 |
| March | 8709 | 11,316 | 110,083 | 7293 |
| April | 5959 | 7743 | 75,321 | 4990 |
| May | 6726 | 8740 | 85,019 | 5632 |
| June | 4498 | 5845 | 56,861 | 3767 |
| July | 5158 | 6702 | 65,199 | 4319 |
| August | 4577 | 5947 | 57,852 | 3833 |
| September | 6334 | 8230 | 80,064 | 5304 |
| October | 5183 | 6735 | 65,518 | 4341 |
| November | 0 | 0 | 0 | 0 |
| December | 2096 | 2724 | 26,496 | 1755 |
| TOTAL | 49,356 | 64,131 | 623,856 | 41,330 |

By comparing the total cost of LPG consumption and the calculated total cost of natural gas, there is an economic saving of 36%, which means that it is cost-effective to replace LPG and switch to the use of natural gas. Figure 10 shows a graphical representation of the total LPG cost and the calculated natural gas cost. It can be concluded that LPG was ahead of natural gas in terms of cost throughout the year by more than one-third of the amount.

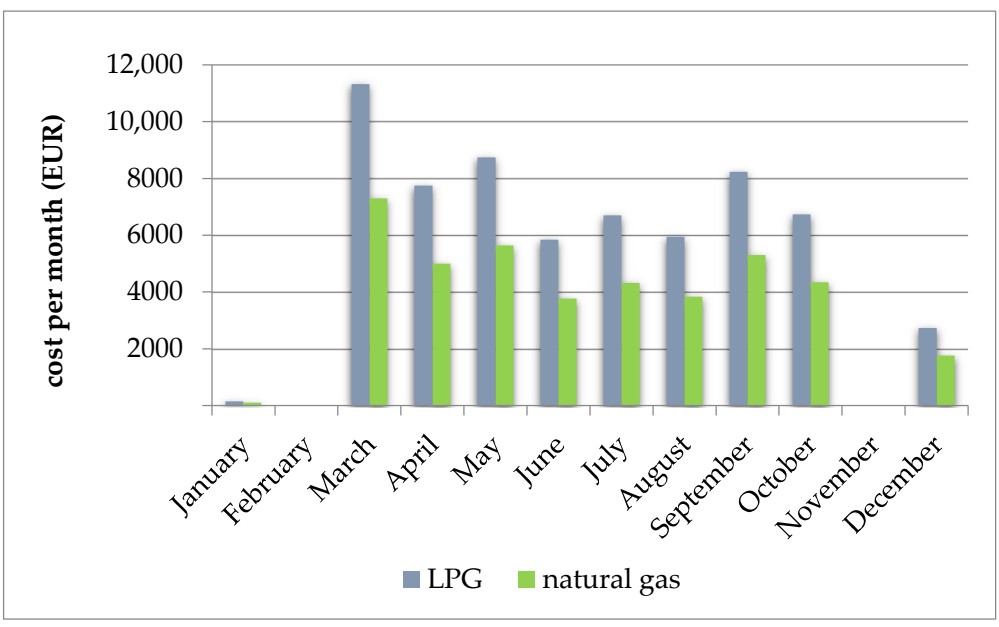

**Figure 10.** Graphical representation of the cost of LPG consumption in 2022 and the calculated cost of natural gas consumption in the same period.

An analysis of gas consumption data in hotels on the Adriatic coast was conducted. The hotels were categorized by size, type of establishment, and location. The results of the analysis showed that medium and large hotels using natural gas consume on average 2.8–3.9 kWh per $m^2$ of gas on an annual basis, while hotels in the same category using LPG consume 0.42–0.56 kWh per $m^2$ of gas. Regression analysis was used to determine the relationship between total annual gas consumption as a function of the total floor area of the hotel building and the total number of rooms. A medium–strong relationship was

found between the above variables, which is a novelty for energy trends in the hospitality and tourism industry in general.

Energy transition in the hotel and tourism industry, in general, is essential for reducing the damaging effects of tourism on the environment. Based on the research findings and significant cost reductions of energy consumption, the promotion of the energy transition can be greatly aided by governments and policymakers through tools such as feed-in tariffs, tax breaks, and subsidies for implementing less carbon-intensive technologies such as the one analysed here, especially for large hotels with bigger energy consumption. However, there are considerable obstacles to the adoption of all renewable energy technologies, including their high cost and the general lack of awareness and information about them, not only in big hotels such as the one explored in this research but also in small hotels and other types of accommodation. Based on the achieved cost savings from the proposed fuel switch, some new investments could be initiated, and recommendations for some of those include window solar film on buildings to reduce heat during summertime and to reduce energy for cooling buildings, installing high-efficiency motors/using variable speed drives for a ventilation system, smart room controls/AC systems in rooms, magnets for windows and motion detectors, solar panels for water heating or electricity production, water pump upgrades with variable speed drives, LED lighting, smart building management systems, implementing geothermal energy for cooling or heating hotel buildings, and using energy (heat) from sewage water for heating hotel buildings.

The results of the analysis are not only applicable to other Croatian tourist regions such as Dalmatia but to all other regions that can switch from LPG to natural gas as a less carbon-intensive fuel and the cost reduction could lead to further investments in renewable fuels as a transition to a carbon-neutral society.

## 5. Conclusions

Based on the data of the hotel's conversion from heating oil to natural gas, significant financial savings of 55.12% and a reduction in energy consumption on an annual basis were found. An analysis of the hotel's potential conversion from LPG to natural gas consumption was conducted. It was calculated that this hotel would reduce total energy consumption costs by 36%.

The results could help the hospitality industry to identify energy reduction measures and cost savings that could be achieved depending on the hotel size and the number of rooms. This could lead to better cost budgeting and investment planning and the planning of the opening and closing of seasonal hotels especially focusing on energy management. For policymakers, this is a direct recommendation to develop feed-in tariffs, tax breaks, subsidies, and other financial measures to encourage the implementation of not only renewable technologies but also less carbon-intensive technologies as a way of reducing carbon footprint.

To continue the research for a more detailed analysis of the gas consumption data, some other regression methods could be used, such as multivariate regression, which shows the relationship of a dependent variable to several independent variables, or clustering, which could pick out different groups of conditions. The dependence of gas consumption on other factors such as the occupancy of the facility, the number of overnight stays, and the area of the room could be observed.

To become one of the top tourist destinations, Croatia could implement the concept of "green hospitality" and use it for marketing purposes. Switching to cleaner fuels could be one of the best options for hotels, as this would undoubtedly lead to lower $CO_2$ emissions without compromising customer comfort. In the future, the introduction of renewable energy sources such as PV and their impact on the economic indicators of hotels could be explored. This would certainly contribute to a complete energy transition towards a sustainable hospitality industry in line with EU targets and the Green Deal.

**Author Contributions:** Conceptualization, M.Š. and D.K.S.; Methodology, M.Š.; Formal Analysis, M.Š. and M.M.; Investigation, M.Š.; Writing—Original Draft Preparation, M.Š., D.K.S. and M.M.; Writing—Review and Editing, M.Š. and D.K.S.; Visualization, M.Š.; Supervision, M.M. All authors have read and agreed to the published version of the manuscript.

**Funding:** This research received no external funding.

**Institutional Review Board Statement:** Hereby we confirm that the manuscript is based on collecting data from human participants representing investigated hotels by questionnaire survey (online survey, email, telephone, and in-person on-site). All investigated subjects were informed that data were collected for scientific research purposes and that results would be published in scientific journals, so they gave their informed consent for inclusion before they participated in the study. All participants were fully informed that anonymity would be assured, that the research was being conducted for the scientific research of the correlation between hotel size and gas consumption, that their data would be used for analytical purposes, and that their data would be anonymous and would not be disclosed in published results of the study. The Study was not performed on humans; therefore, there was no need for an approved protocol according to the Declaration of Helsinki. The study was performed according to the Ethical Codex of the University of Zagreb (2009) downloadable at http://www.unizg.hr/fileadmin/rektorat/O_Sveucilistu/Dokumenti_javnost/ Propisi/Pravilnici/Eticki_kodeks.pdf (accessed on 10 May 2023) with no requirements for Ethics Committee or Institutional Review Board.

**Informed Consent Statement:** Informed consent was obtained from all subjects involved in the study.

**Data Availability Statement:** The raw data of this work will be provided on reasonable request by contacting the corresponding author.

**Conflicts of Interest:** The authors declare no conflict of interest.

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
