# Peer review of "Correlations between Hotel Size and Gas Consumption with a Feasibility Analysis of a Fuel Switch—A Coastal Case Study Croatia Adriatic"

_sustainability, doi:10.3390/su15118595_

Round 1

Reviewer 1 Report

The article "Correlations between hotel size and gas consumption with a  feasibility analysis of a fuel switch – a coastal case study Croatia Adriatic " is survey-based. The overall study is good and seems to be a valuable addition.

-          Introduction section is weak. Needs to improve it significantly

-          Adds more literature. There is a lack of literature. It would be good if the authors add more literature in the introduction part or as a separate section. It would strengthen the introduction section as the introduction section needs to improve.

-          Second,  the study hypothesis section also can be added

-          For analysis, the authors used graphical correlation analysis. Why not some other advanced methods? However, this method is also presenting a clear picture. But if any other advanced method is possible, it would be a good addition. As robustness analysis, it can be included.

-     Conclusion section needs to improve, especially Policy recommendation needs to improve

Overall it is a good study. There is a need to improve the paper structure, add literature, and some advanced methods. 

fine. maybe some minor issues

Author Response

Dear Reviewer and Editor,

Thank you very much for your comments and recommendations for our manuscript “Correlations between hotel size and gas consumption with a feasibility analysis of a fuel switch – a coastal case study Croatia Adriatic”.  We have tried to address all the comments and therefore the whole text has been changed in the hope that you would consider it for publishing in Special Issue" Sustainable Solutions for Carbon Dioxide Emissions Mitigation through Utilization Schemes". Please find point by point answers attached in file cover_letter for editor and reviewer1. 

Reviewer 2 Report

Dear/s Author/sy,

Re: Manuscript “Correlations between hotel size and gas consumption with a feasibility analysis of a fuel switch – a coastal case study Croatia Adriatic”

Reviewer’s report:

The objective of the paper is relevant as it is to analyze the consumption of hotel accommodations to try to help hotels reduce their energy bill and contribute to the protection of the environment. The introduction lacks more recent literature. It is necessary for the authors to search scientific databases such as the JCR to update the literature. A simple keyword search provides countless results in the last 5 years (energy transition - more than 100,000 articles -, gas consumption - more than 24,000 -, green hospitality -371-). This new literature review should make it possible to better highlight the existing research gap. Likewise, the discussion has to be a section in which the literature of the theoretical framework is compared with the results, but it cannot be a continuation of these. Also, the conclusions must be shorter and in relation to the objective formulated. In them, figures cannot appear.

Best regards

Author Response

Dear Reviewer and Editor,

Thank you very much for your comments and recommendations for our manuscript “Correlations between hotel size and gas consumption with a feasibility analysis of a fuel switch – a coastal case study Croatia Adriatic”.  We have tried to address all the comments and therefore the whole text has been changed in the hope that you would consider it for publishing in Special Issue" Sustainable Solutions for Carbon Dioxide Emissions Mitigation through Utilization Schemes". Please find point by point answers attached in file cover_letter for editor and reviewer2.

Reviewer 3 Report

Dear authors,

I have been invited to review your paper. Attached you will find some detailed comments inserted directly in the PDF file. In my view, the submission does not qualify for publication in Sustainability.

The submission lacks a comprehensive and complete literature review. The objectives and novelty as well as the contributions to the existing body of knowledge are not clearly spelt out or are limited. The methodology used is not properly introduced and the policy recommendations are not sustained by the analysis undertaken.

Overall, the submission contains a case study with limited interest for a wide audience such as that of Sustainability.

Best regards,

Anonymous reviewer

Author Response

Dear Reviewer and Editor,

Thank you very much for your comments and recommendations for our manuscript “Correlations between hotel size and gas consumption with a feasibility analysis of a fuel switch – a coastal case study Croatia Adriatic”.  We have tried to address all the comments and therefore the whole text has been changed in the hope that you would consider it for publishing in Special Issue" Sustainable Solutions for Carbon Dioxide Emissions Mitigation through Utilization Schemes". Please find point by point answers attached in file cover_letter for editor and reviewer3.

Round 2

Reviewer 1 Report

Accept

Reviewer 2 Report

Dear/s Author/s,

Re: Manuscript “Correlations between hotel size and gas consumption with a feasibility analysis of a fuel switch – a coastal case study Croatia Adriatic

Reviewer’s report:

The bibliographic references have been significantly updated and the authors present a fairly new topic. My opinion is in favor of publication of the paper.

* Personally, I have understood the entire exposition in English of the paper, but I am not a native English speaker. I consider it appropriate that someone from your journal check the writing in this language.

Best regards

Reviewer 3 Report

Dear authors,

Thank you for your responses to the questions raised by the reviewers and for the revised version of your paper.

Best regards,

Anonymous reviewer

Dear authors,

I would like to suggest that the text is proofread by a professional as I have spotted a couple of grammar mistakes.

Best regards,

Anonymous reviewer